# *Lucilia silvarum* Meigen (Diptera: Calliphoridae) Is a Primary Colonizer of Domestic Cats (*Felis catus*)

**DOI:** 10.3390/insects15010032

**Published:** 2024-01-04

**Authors:** Kelly Bagsby, Krystal Hans

**Affiliations:** 1Department of Entomology, Purdue University, 901 W State St., West Lafayette, IN 47907, USA; hans3@purdue.edu; 2College of Veterinary Medicine and College of Medicine, University of Florida, Gainesville, FL 32611, USA

**Keywords:** entomology, forensics, veterinary, feline, forensic entomology, Indiana, Calliphoridae, animal cruelty

## Abstract

**Simple Summary:**

This study analyzed the initial blow fly colonizers of cats in August 2021 in Indiana. *Lucilia silvarum* was a primary colonizer of cats in this study, which was a surprising and unexpected finding. *Lucilia silvarum* has been referred to as a blow fly species that only performs myiasis on amphibians up until recently. In 2014, another blow fly species, *Lucilia bufonivora,* was discovered in established collections in North America. With this discovery, further reexaminations of fly specimens found that *L. bufonivora* has been in North America since the 1950s and that the specimens identified as *L. silvarum* were actually *L. bufonivora*. Given the history of *L. silvarum* and that this fly appeared as a primary colonizer of cats, the purpose of this research is to compile all the *L. silvarum* literature while bringing awareness to the forensic entomology community that this blow fly is forensically relevant. The implications of *L. silvaurm* being a primary colonizer will aid future investigations of animal abuse or cruelty. Correct identification of blow fly species is paramount when calculating time of colonization estimates. These estimates aid investigators in determining timelines regarding wounds or death.

**Abstract:**

*Lucilia silvarum* Meigen (Diptera: Calliphoridae) is widespread throughout North America and Europe. Described in 1826, this blow fly was quickly associated with myiasis in amphibians, and to date has rarely been reported in carrion. There is limited data regarding the time of colonization of animals with fur and the interpretation of this data is difficult due to variation in the animal models used. During an examination of initial insect colonization of cats (*Felis catus*) with light and dark fur, twelve domestic short-haired cats were placed in cages 15.2 m apart in a grassy field in West Lafayette, Indiana, USA. Eggs from initial oviposition events were collected and reared to identify the colonizing species. Three species of *Lucilia* (Diptera: Calliphoridae), including *L. silvarum*, colonized the cats on the initial day of placement. In this study, *L. silvarum* was the primary colonizer of cats, and this may be the first study where a large number of *L. silvarum* were collected. Further studies should include development studies on *L. silvarum* to understand its life history and aid in time of colonization estimations. More work regarding the colonization of furred mammals is needed to further examine *L. silvarum* as a primary colonizer.

## 1. Introduction

Forensic entomology is the study of insects that answers questions within the judicial system [1]. Insects collected as evidence can provide information such as when a person or animal died, when the decedent was available for colonization, and if they were moved, and can potentially provide other information about the death such as about wounds [1]. Forensic entomologists are interested in development data because the most common analysis requested by law enforcement is an estimate of time since death [2]. Understanding how a blow fly develops and how long each life stage lasts can aid in calculating the time of colonization (TOC). Time of colonization is the period of time between blow fly oviposition, subsequent feeding, and the discovery of the remains, which can be calculated using the development data from laboratory studies [3]. Variables such as temperature [4], type of tissue [5,6,7], alternative food sources [8], photoperiod [9,10], and coexistence with other species [11,12,13] have been found to affect the development of blow fly larvae. Understanding all of these variables is necessary because of the effect they have on the development of blow fly larvae. In addition to development data, there are established minimum and maximum threshold temperatures, which are experimentally determined [4,6,14]. Experimentally determined development data can be used by forensic entomologists to calculate the TOC. Awareness of blow fly life stage development under set conditions can aid forensic entomologists in understanding how environmental variables affect oviposition and development.

Understanding oviposition behavior, life cycle development, and the species preferences of blow flies (Diptera: Calliphoridae) is important for the field of forensic entomology. Oviposition behavior can be affected by temperature [15], sunlight [16], volatile organic compounds (VOCs) [17], and stage of decomposition [15], in addition to how accessible remains are for colonization [18]. Blow fly adults and larvae are poikilothermic, meaning that ambient temperature affects their growth and development [15]. Extreme temperatures, either above or below the insect’s temperature threshold, can speed up, slow, or stop development. Sunlight can affect the microhabitat where remains are located, resulting in higher temperatures that accelerate the decomposition process [19]. Sunlight has also been shown to affect which blow fly species oviposit on remains [16,20]. As soon as death occurs, the body begins to break down, which results in the production of VOCs. Blow flies, through their olfactory system, are able to identify these chemical cues and locate the source of these odors, the decomposing remains [17]. Although understanding VOCs and the implications of attractions need further research, it is thought that the different molecules released during the decomposition stages can affect insect behavior [21]. Sulfur-based molecules may influence the attraction of flies, and ammonium-rich molecules may promote egg laying [22,23,24,25,26]. Visual stimuli is thought to be another important cue used to supplement olfactory cues when locating decomposing remains [18,27]. The remains constantly change as the tissues decompose. These changes are reflected by the types of insects that arrive, feed, and then leave in waves [15]. Blow flies can arrive at remains almost immediately after exposure when the remains are still fresh, whereas beetles arrive at remains that have been decomposing for a longer time resulting in dryer conditions [15]. Another variable that may affect oviposition is accessibility. Oviposition can be delayed for a number of reasons, including whether the insects have access to the remains, such as inside dwellings or vehicles with closed windows and doors [28]. Remains can also be concealed through burial or by wrapping with fabric or plastics, which may delay arrival and subsequent colonization [28].

In forensic entomology field studies, the animal model most commonly used is a pig [29,30,31]. Acceptance of pigs as the human equivalent for entomology research is due to their similar body mass, skin thickness, hair coverage, microbiome, and insect colonization and subsequent succession of seres [29,30,32,33]. In North America, other animal species have been used throughout the history of forensic entomology and decomposition including dogs, cats, foxes, squirrels, and more (Table 1). Cat decomposition has historically been researched very little, with only two studies conducted since 1975 and one case review in 2015 [34,35,36]. Using other animal species in a decomposition study adds to the knowledge gap around animals with fur. Animal fur has been shown to affect patterns of decomposition, the physical characteristics of known stages of decomposition, and access to the tissues.

Additionally, some blow flies have preferences for what type of animal species or what type of decaying material [15,66] they colonize. Common names can denote this preference, such as the toad fly [18,67]. Preference towards certain animal species may result in myiasis, which occurs when live animals are infested with dipterous larvae [68,69,70]. The larvae feed on the host’s living and/or necrotic tissues, bodily fluids, or ingested food [68,69,70]. *Lucilia bufonivora* (Moniez) is commonly known as the toad fly and has been known to perform obligate myiasis on amphibians, with its primary host being the common toad (*Bufo bufo*) followed by other frogs, toads, and salamanders [50,67,71]. In the Palearctic regions, *L. bufonivora* is known to lay its eggs on the back and flanks of amphibians and then, after hatching, the maggots migrate towards the eyes and nasal cavities [48,72]. Up until 2019, *Lucilia silvarum* (Meigen) had been documented to primarily colonize live amphibians [72]. In cases involving *L. silvarum* within the Nearctic region, eggs are typically laid on the back and then migrate to other regions of the frog’s body to begin feeding [48,49,73]. After hatching, maggots tend to migrate towards to the head [74,75] and then may begin to feed and burrow into the head cavity reaching the parotid gland [48,74], eyes [75], and ears [75]. Other locations with lesions where maggots are found include the neck [48], back [76], legs [48,50], flanks [49,50,77], and abdomen [49].

### 1.1. The History of Lucilia silvarum

*Lucilia silvarum* is a species of blow fly of the *Lucilia* genus which was described by Meigen in 1826 and has an interesting history (Figure 1). The *Lucilia* genus is collectively known as the green bottle blow fly due to its metallic green coloration and contains twelve species in North America [78]. *Lucilia silvarum* is known to inhabit the holarctic region, which is the Neartic and Palearctic regions combined, covering the northern continents [18,69,78]. In North America, *L. silvarum* is considered to be widespread and has been documented to occur as far north as British Columbia, Canada, and as far south as southern Florida, USA [78]. *Lucilia silvarum* was first documented performing myiasis on amphibians in 1891 and has been associated with amphibian myiasis, with reports of eggs being laid on the back, neck, legs, and parotid glands [68,72,79].

The first recorded observation of amphibian myiasis was documented in 1865 [68]. In 1870, other scientists in Holland and Luxembourg described that amphibian myiasis commonly occurs during the August and September months and results in the death of the amphibians. In 1876, Moniez reared these adults and described the adult fly as *L. bufonivora*. In 1891, Dunker also reared blow fly larvae from a case of amphibian myiasis, where he identified these adults as *L. silvarum* [68]. Beginning in the mid-1940s, authors like Hall and Zumpt cautioned that there seems to be confusion regarding *L. silvarum* performing amphibian myiasis [68,80]. In 1948, Hall stated that *L. silvarum* is rarely collected from decomposing substances and is not attracted to baits containing meat [80]. Additionally, Hall stated that records of amphibian myiasis in the Palearctic region are due to either *L. silvarum* or *L. bufonivora* and that authors had not distinguished between these two species in any publications [80]. In 1965, Zumpt claimed that Dunker misidentified these flies as *L. silvarum* and believed that the flies were actually *L. bufonivora* [68]. Ever since this identification, several authors have associated *L. silvarum* with amphibian myiasis, which has been an ongoing issue affecting many publications. A contributing factor to this confusion and misidentification could be due to the fact that *L. bufonivora* was not included in any identification keys in North America until Jones et al., 2019 [78,79,80,81].

Many publications throughout the years have mentioned the collection and identification of *L. silvarum*. These publications consist of fly surveys and incidental findings of oviposition on mammals and birds, which seem to have contradicted the idea that *L. silvarum* is solely parasitic towards amphibians. A fly survey conducted in Michigan, USA, in 1948–1950 resulted in 11.9% to 27.8% of the flies being identified as *L. silvarum*, which indicates that the *L. silvarum* population was sustainable and greater than what could be supported solely by amphibian myiasis [69]. In Finland, researchers were also conducting surveys and analyzing competition between species [11]. In the wild, the *L. silvarum* population was close to the population of *L. illustris*, at 31.8% and 34.0%, respectively. In Indiana, a survey using beef liver was conducted, and *L. silvarum* accounted for 24.7% of the total calliphorids collected [82]. Davies analyzed blow fly species differences between small and large carcasses, mice, and sheep, respectively [83]. *Lucilia silvarum* only colonized the mice. These surveys show that the wild *L. silvarum* population, in a variety of locations, is typically below 32%.

Dating back to 1951, there have been a few cases of incidental findings that involved colonization by *L. silvarum* on mammals. In Wisconsin, a case of possible myiasis on an apparently healthy rat was discovered [69]. *Lucilia silvarum* accounted for 65% of the identified blow flies, which contradicts previous knowledge regarding the biology of *L. silvarum* [69]. An incidental finding of *L. silvarum* oviposition on a duck carcass was also discovered in California [84]. Eggs were reared and the flies were identified as *L. silvarum* and *L. sericata*. Upon this discovery, in 1968, a fly survey was conducted where *L. silvarum* accounted for 17.1% of the collected population [84]. The area where the duck carcass was discovered is a habitat with saline conditions, which is not a hospitable habitat for amphibians, and, since there were no amphibians in this area, *L. silvarum* was an unexpected species to be found given the assumption that this species only performed myiasis on amphibians. These two cases involving the colonization of *L. silvarum* show that this species may prefer animals small in size or prefer resources with low competition.

**Figure 1 insects-15-00032-f001:**
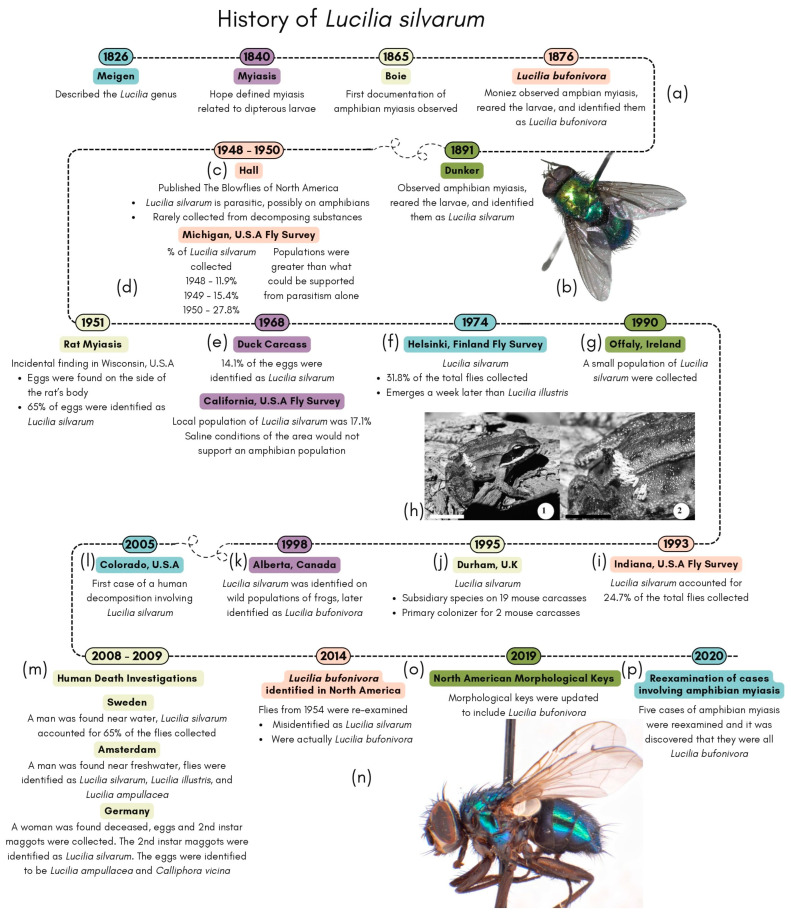
The history of *Lucilia silvarum* and other notable discoveries: (**a**) Zumpt (1965) provides a summary of early discoveries involving *L. silvarum* and *L. bufonivora* [68]; (**b**) *Lucilia* silvarum; (**c**) Hall (1948) published The Blowflies of North America and provided noteworthy findings regarding *L. silvarum* [80]; (**d**) CDC survey and incidental *L. silvarum* finding by Dodge (1952) [69]; (**e**) incidental *L. silvarum* finding and subsequent survey conducted by Brothers (1968) [84]; (**f**) Finland blow fly study conducted by Hanski (1974) [11]; (**g**) Blackith and Blackith (1990) conducted a survey analyzing small corpses [85]; (**h**) image with eggs on a frog’s back, reproduced with permission from Bolek and Janovy, Observations on Myiasis by the Calliphorids, *Bufolucilia silvarum* and *Bufolucilia elongata*, in Wood Frogs, *Rana sylvatica*, from Southeastern Wisconsin; published by KGL Publishing Services, 2004 [49]; (**i**) Haskell (1993) conducted a fly survey in Indiana [82]; (**j**) a study by Davies analyzed small and large carcasses (1999) [83]; (**k**) Eaton et al. (2008) observed amphibian myiasis [50]; (**l**) *Lucilia silvarum* was identified in a human death investigation [86]; (**m**) three human death investigations were associated with *L. silvarum* [87]; (**n**) image of *Lucilia bufonivora*, reproduced with permission from Tantawi and Whitworth, First Record of *Lucilia bufonivora* Moniez, 1876 (Diptera: Calliphoridae), from North America and a key to North American species of the *L. bufonivora* species group; published by Zootaxa, 2014 [88]; (**o**) Jones et al. (2019) published an influential key which included *L. bufonivora* [78]; (**p**) amphibian myiasis cases were re-examined by Whitworth et al. (2021) [72].

### 1.2. The Misidentification of Lucilia silvarum and the Reexamination of Amphibian Myiasis Cases

In 2014, *L. bufonivora* was discovered in the Canadian National Collection of Insects [88]. These flies were found to be originally identified as *L. silvarum*, which turned out to be a misidentification. The specimens that were re-examined were collected all over Canada from British Columbia to Manitoba dating back to 1954. Upon this discovery, a key was made to distinguish between *L. bufonivora*, *L. silvarum*, and *Lucilia elongata* (Hough) [88]. Prior to 2014, it was thought that *L. bufonivora* was strictly Palaearctic. In 2019, an important key for the calliphorids of North America was published by Jones et al., which is well known and readily used throughout the entomology field [78]. After this finding, flies were reexamined from Calling Lake in Boreal Alberta, Canada, and Pine Lake in Alberta, Canada, that were originally identified as *L. silvarum* [89]. Through phylogenetic analyses, they were identified as *L. bufonivora* [89]. An article published in 2021 reexamined three cases [48,49,76] of flies associated with amphibian myiasis in Wisconsin, USA [72]. Species were identified using the 2014 and 2019 updated morphological keys [78,88] and confirmation was done by an expert in blow fly taxonomy by examining the male and female terminalia [81]. The reexamined flies were all found to be misidentified as *L. silvarum* and were actually all *L. bufonivora* [72].

*Lucilia silvarum* and *L. bufonivora*’s morphological features are similar and only differ in small features, the color of their calpyters, and the number of post-sutural acrostichal bristles present [78] (Table 2). In addition to their morphological features, phylogenetic analysis has demonstrated that these two flies are sister species [67,89]. These similarities could have been a contributing factor leading to the misidentifications of *L. silvarum* and *L. bufonivora*.

### 1.3. Previous Studies Involving Cats

*Lucilia silvarum* had not been collected and identified from cats in North America until this study. This species had been collected previously in Indiana, but was associated with pigs [24,82]. Johnson analyzed the seasonal variations in Illinois during the months of June 1968 to October 1969 using a wide variety of small furred mammal carcasses (*n* = 39) [34]. A single cat accounted for one of the animal models used. The fly species collected were categorized into different seasons, spring, summer, and fall, but did not categorize by animal model. In this study, *Phormia regina* (Meigen) and *Calliphora vicina* (Robineau-Desvoidy) were collected in all three seasons and *Lucilia sericata* (Meigen) was collected only during the summer and fall. Johnson also noted that flies of the Sarcophagidae family were found during the months of August and September but that those flies played a minor role. A second study, conducted by Early and Goff, examined decomposition and arthropod succession on cat carcasses during the winter (October to December) inside Diamond Head Crater and during the summer (March to May) at the University of Hawaii in Manoa [35]. *Chrysomya rufifacies* (Maquart), *Chrysomya Megacephala* (Fabricius), and flies of the Sarcophagidae family were collected during both seasons, while *Lucilia cuprina* (Wiedemann) was only collected during the summer months. The composition of flies collected at the Manao site was primarily flies that live close to humans and dwellings, whereas the Diamond Head site had more diversity among blow fly assemblages. The last study mentioning cats, by Sanford, completed a review of scene photography from cases involving human decedents and their pets from 2009 to 2014 [36]. Only three cases included forensic entomology as evidence. In one case, *P. regina* was identified with collections from the cat. In another case, flies identified from cat remains included the families Phoridae and Sarcophagidae. Since the current study is the first using cats as the animal model since 1986, identifying three new species in the colonization of cats is a noteworthy finding.

The purpose of this study was to analyze the relative abundance of blow flies that colonized cats with light and dark fur. Understanding which blow fly species arrive to colonize cats is a significant contribution to the forensic entomology field, especially when investigating claims of animal death, neglect, or abuse. Previous research has shown which fly species are typically collected in Indiana and which species have been identified to colonize cats. Initially, this study examined the decomposition and diversity of insects arriving and colonizing domestic cat (*Felis catus*) carcasses with light and dark fur. When examining the insects collected during the study, the primary blow fly colonizer was *L. silvarum*. Based on previous studies and historical literature on this species, this was a surprising finding, which required a closer examination of the history of *L. silvarum* and the results of this study.

## 2. Materials and Methods

Methods for this study were described in detail in Bagsby et al. (in press) but will be summarized briefly below [90]. This study was conducted in August of 2021 in Indiana, USA, using twelve domestic short-haired cats: three light cats and six dark cats with a mean weight of 5.3 +/− 0.11 kg. The cats were ethically sourced from a local shelter after they were chemically euthanized via cardiac stick with Euthasol, containing the active ingredients pentobarbital sodium and phenytoin sodium, according to the animal control protocol, for reasons unrelated to this study. On the initial day of this study, the cat carcasses were photographed, and initial documentation included fur color and weight (kg). Once documentation was completed, the cats were placed on their left side directly on the ground, 50 m apart, protected with a cage, with their head facing north, in an open field chosen randomly at Purdue University’s Forensic Entomology Research Site (40.426734, −86.949390) [91].

The area of this study was an Eastern Temperate Forests ecoregion, more specifically the Central USA Plain ecoregion [92]. The climate of this area is a humid continental climate, with warm summers and cold winters; all four seasons are well represented [93]. The research site was an open grassy field with trees surrounding the edges that allowed for equal sun and shade exposure. Precipitation data were collected from the Lafayette Purdue University Airport weather station (40.4124, −86.9474) via the National Oceanic and Atmospheric Administration (NOAA). Temperature and relative humidity were collected hourly using data loggers (HOBO MX2300, Onset Computer Corporation, Bourne, MA). To account for any variability, the field temperature data were corrected using a certified weather station. A linear regression was completed using the averaged HOBO unit temperature data and the NOAA weather station temperature data [94,95,96]. The mean temperature of the study was 35.51(°C) +/− 0.6133, mean relative humidity was 67.11% +/− 0.8254, and there was a total of 1.97 inches of rain.

Materials and methods from Brundage and Byrd were followed to document insect activity, collection techniques, and rearing protocols [2]. On the initial day of the study, the cats were observed for initial insect arrival and colonization every two hours, for a total of twelve hours, twice a day until day six, and then once a day thereafter. During each observation, data were collected regarding the time to first oviposition event, subsequent new oviposition events, and oviposition sites selected for colonization. A total of 38 samplings occurred on the initial day of the study. Once all of the eggs from the field emerged as adults, they were identified using a morphological key [78]. A selection of adult flies from this data set were confirmed by a blow fly taxonomist (T. Whitworth, personal communication).

### Statistical Analysis

Relative abundance was calculated for each species to determine the diversity of the blow flies that colonized light and dark cat carcasses. Data were normally distributed (Shapiro–Wilk, *p* > 0.05). A two-tailed paired *t*-test was performed to compare the number of fly species that colonized light-fur cats and dark-fur cats.

## 3. Results

A total of 145 flies emerged from the egg samples collected from the cat carcasses on the initial day of the study. The flies that colonized the cats were *Lucilia coeruleiviridis* (Maquart), *L. silvarum*, *Lucilia illustris* (Meigen), and flies in the Sarcophagidae family. Of the 38 samples collected in this study, 57% of the samples failed to develop. Relative abundance calculations showed that *L. silvarum* was the primary colonizer of cats with 58.62% relative abundance, followed by *L. coeruleiviridis* (37.24%), Sarcophagidae (2.76%), and *L. illustris* (1.38%). There was no significant difference in fly species colonization between fur colors (t = 1.438, df = 3, *p* = 0.2461). Both light and dark cats were colonized by *L. coeruleiviridis* and *L. silvarum*; *L. illustris* only colonized light cats and Sarcophagidae only colonized dark cats.

## 4. Discussion

This study provides information about the fly species that colonize cat carcasses in Indiana, which is a significant addition to the field of forensic entomology because *L. silvarum* had not been collected or identified in previous studies using cats as the animal model [34,35,36]. Additionally, *L. silvarum* is not a common blow fly species encountered in death investigations or research in Indiana [97,98]. *Lucilia silvarum* had also rarely been collected in other states within the United States [48,49,69,84,86] or in Europe, with only three reports from human death investigations [11,87].

Most of the flies identified from this study were *L. silvarum*, which was an unexpected result, especially with it being the primary colonizer in this study. Although *L. coeruleiviridis* was also collected in this study, the number of *L. coeruleiviridis* could be underrepresented because this species of fly is incredibly difficult to rear in laboratory settings and many egg samples collected from the field failed to develop and reach adulthood [99]. Previous research from studies in Indiana shows that the total number of *L. silvarum* collected is typically lower than 25% [82]. In other studies across the United States, the prevalence of *L. silvarum* ranged from as low as 11.9% to as high as 65% [69,84]. Most of the reference material prior to 2021 regarding *L. silvarum* associates this species with amphibian myiasis, but new research has shown that these identifications prior to 2014 were incorrect [67,72,77,89]. Given these misidentifications, the source material for *L. silvarum* prior to 2021 should be reevaluated and used with caution.

*Lucilia silvarum* may have been a primary colonizer of cats because it is a species that prefers carcasses of smaller sizes or prefers to oviposit when there is a lack of competition. Carrion are an ephemeral resource, and the size of carrion may affect blow fly species’ assemblages and subsequent survival [3,83,85]. Davies analyzed seasonal and spatial changes between small and large carcasses and found that *L. silvarum* was a primary colonizer in two mice in July 1997 and a subsidiary species in 19 other mice but did not colonize any of the sheep [83]. For one mouse, a total of 76 *L. silvarum* were collected and identified, for a total of 57%. For the other mouse, there was a total of 124 *L. silvarum*, for a total of 98%. When *L. silvarum* was a subsidiary species, the prevalence ranged from 0.1% to 43% [83]. In 1951, an incidental finding of myiasis on a rat in Wisconsin indicated that 65% of the reared flies were *L. silvarum* [69]. Another incidental finding in 1968 occurred in California when a duck carcass was found with 15% of the flies identified as *L. silvarum* [84]. With there only being four blow fly species to colonize the cat carcasses in this study, *L. silvarum* may prefer to oviposit when there is a lack of competition from other blow flies. Since carrion are ephemeral, competition between species can be significant [11,13,28]. A study using small pigs, weighing 5.1 kg, resulted in a total of two *L. silvarum* out of 10,723 total Calliphoridae flies collected; its prevalence was lacking at just 0.01% [16]. When there was not a diverse population of blow flies that colonized remains, *L. silvarum* was a primary colonizer [69,83,84]. Competition may be so extreme that a single or few species may dominate an area [11]. A study analyzing competition among carrion flies in Finland showed that *L. silvarum* was outcompeted when coexisting with *L. illustris* but was able to maintain its population when caged alone [11]. Another finding in the Finland study showed that *L. illustris* emerged a week earlier, which could give *L. illustris* a competitive advantage over *L. silvarum* [11]. In Indiana, *P. regina* and *L. sericata* are the dominant species to use in research and colonize remains [98]. Since other blow fly species are dominant in the area of this study, *L. silvarum* may take advantage of resources early on in the decompositional process before other species arrive or display preferential colonization of small carcasses or certain animal species—specifically cats.

Discovering that *L. silvarum* was a primary colonizer of cats in Indiana was a significant finding for the forensic entomology community. The previous knowledge that the forensic entomology field had about *L. silvarum* was that this fly species was parasitic towards amphibians by performing myiasis. It was not until 2014, when *L. bufonivora* specimens were discovered in a North American collection, that a close look into these two species occurred [88]. This discovery prompted the reexamination of other specimens in the collection, which resulted in the revelation that the *L. silvarum* specimens, dating back to 1954, were misidentified [72]. Given these rediscoveries, in addition to the identification of *L. silvarum* on cat carcasses, understanding this species more in depth is necessary. Awareness of how cats and other animals with fur are colonized by blow flies is an important aspect of determining time of colonization estimates of injuries or death. The implications of these findings will aid in future investigations of animal cruelty and abuse involving entomological evidence. Since time of colonization estimates rely on correct species identification, development studies need to be conducted for *L. silvarum*. Limited data exist in the forensic science community about decomposition, oviposition behavior, and forensically relevant blow fly species for animals with fur; therefore, any contributions to closing this knowledge gap will be beneficial.

## Figures and Tables

**Table 1 insects-15-00032-t001:** Studies involving blow fly oviposition of different animal models in North America. Swine are the dominant animal model used regularly in forensic entomology studies, followed by mice/rats. There is a need for the use of other animal models to have a comprehensive understanding of how blow fly oviposition behavior is affected by fur, scales, feathers, and different skin types.

Animal Model	Author, Year	Location	Season
Alligator	Watson and Carlton, 2003 [37]	Louisiana	Spring
	Watson and Carlton, 2005 [38]	Louisiana	Winter, Fall
	Nelder et al., 2009 [39]	Alabama	Spring
Bear	Watson and Carlton, 2003 [37]	Louisiana	Spring
	Watson and Carlton, 2005 [38]	Louisiana	Winter, Fall
	Swiger et al., 2014 [40]	Florida	Summer
Bird	Lord and Burger, 1984 [41]	New Hampshire	Spring, Summer, Fall
	Bennett and Whitworth, 1991 [42]	Ontario, Canada	Summer
	Sawyer et al., 2022 [43]	Texas	Winter, Summer
Cat	Johnson, 1975 [34]	Illinois	Spring, Summer, Fall
	Early and Goff, 1986 [35]	Hawaii	Winter, Summer
	Sanford, 2015 [36]	Texas	Spring
Bobcat	Richards et al., 2015 [44]	Florida	Fall
Chipmunk	Payne, 1965 [33]	South Carolina	Spring, Summer
Deer	De Jong, 1994 [45]	Colorado	
	Watson and Carlton, 2003 [37]	Louisiana	Spring
	Watson and Carlton, 2005 [38]	Louisiana	Winter, Fall
	Cammack and Nelder, 2010 [46]	South Carolina	Fall
Dog	Reed, 1958 [47]	Tennessee	Spring
	De Jong, 1994 [45]	Colorado	
	Sanford, 2015 [36]	Texas	Spring
Coyote	Richards et al., 2015 [44]	Florida	Winter
Frog/Toad	Payne, 1965 [33]	South Carolina	Spring, Summer
	Bolek and Coggins, 2002 [48]	Wisconsin	Spring, Summer
	Bolek and Janovy, 2004 [49]	Wisconsin	Summer
	Eaton et al., 2008 [50]	Alberta, Canada	Spring, Sumer
Mouse/Rat	Payne, 1965 [33]	South Carolina	Spring, Summer
	Greenberg, 1990 [51]	Illinois	Summer
	De Jong, 1994 [45]	Colorado	
	Patrician and Vaidyanathan, 1995 [52]	New York	Fall
	Tomberlin and Adler, 1998 [53]	South Carolina	Winter, Summer
	De Jong and Hoback, 2006 [54]	Colorado	Summer
	Sawyer et al., 2022 [43]	Texas	Winter, Summer
Opossum	Johnson, 1975 [34]	Illinois	Spring, Summer, Fall
	Richards et al., 2015 [44]	Florida	Summer
Rabbit	Dautaras et al., 2018 [29]	Tennessee	Winter, Spring, Summer
	Denno and Cothram, 1975 [55]	California	Winter, Spring, Summer, Fall
	Johnson, 1975 [34]	Illinois	Spring, Summer, Fall
	De Jong and Chadwich, 1999 [56]	Colorado	Summer
Racoon	De Jong, 1994 [45]	Colorado	
	Joy et al., 2002 [57]	West Virginia	Spring
Shrew	Payne, 1965 [33]	South Carolina	Spring, Summer
Skunk	De Jong, 1994 [45]	Colorado	
Swine	Payne, 1965 [33]	South Carolina	Spring, Summer
	Watson and Carlton, 2003 [37]	Louisiana	Spring
	Watson and Carlton, 2005 [38]	Louisiana	Winter, Fall
	Slone and Gruner, 2007 [58]	Indiana	Summer, Fall
	Slone and Gruner, 2007 [58]	Florida	Winter, Spring
	Bugajski et al., 2011 [59]	Indiana	Summer
	Bugajski and Tolle, 2014 [60]	Indiana	Fall
	Mohr and Tomberlin, 2014 [31]	Texas	Winter, Summer
	Zurawski et al., 2014 [61]	Michigan	Summer
	Weidner et al., 2016 [62]	New Jersey	Summer
	Dautaras et al., 2018 [29]	Tennessee	Winter, Spring, Summer
	Mañas-Jordá et al., 2018 [63]	Mexico	Summer, Fall
	Matuszewski et al., 2019 [30]		
Turtle	Ambercrombie, 1977 [64]	Maryland	Fall
	Abell et al., 1982 [65]	Massachusetts	Summer
	De Jong, 1994 [45]	Colorado	

**Table 2 insects-15-00032-t002:** Similarities and differences among *Lucilia silvarum* and *Lucilia bufonivora*.

*Lucilia silvarum* ^1^	*Lucilia bufonivora* ^1^
Subcostal sclerite without setae	
Black basicosta	
Palp black or brown	
Two postsutural intra-alar setae	
Presutural intra-alar seta present	
Male upper calypter pale, lower calypter tan	Male upper and lower calypter pale
Three postsutural acrostichal bristles	Two postsutural acrostichal bristles

^1^ Jones et al., 2019 [78].

## Data Availability

The data presented in this study are available upon request from the corresponding author.

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
