# Peer review of "Lucilia silvarum Meigen (Diptera: Calliphoridae) Is a Primary Colonizer of Domestic Cats (Felis catus)"

_insects, 2024, doi:10.3390/insects15010032_

Round 1

Reviewer 1 Report

Comments and Suggestions for Authors

Lines 40-42 The structure of this sentence is quite confusing; we suggest to rephrase is and maybe break it down in multiple sentences to make it clearer. 

From Line 42 The authors talk about the importance of development data but they do not say what those are, how those are importance in the estimation of time of death. Even in this case clearer and more specific sentences would be more beneficial, especially considering that the journal is not a forensic entomology one and it has a broader audience

HISTORY Of L. SILVARUM

Are there any information on how L. silvarum was identified in the studies mentioned in this section? Was it a morphological identification? If yes, which keys were used for it? I know the identification is discussed better in the next section, but it would help to cite them here as well 

Line 199 Lucilia silvarum should be italicized; same in Fig. 1

Line 294 and 296 Lucilia silvarum should be italicized

Line 312 What does citation 116 refer to in this sentence? If it’s the Bagsby citation maybe it should be after the source is mentioned in the sentence. Also, if it is in revision, maybe the journal where it was submitted to should not be mentioned.

METHODS

Since only oviposition events were considered for sampling, how many samplings actually occurred and on what experimental days? 

STATISTICAL ANALYSES

Was a normality test performed?

Line 376 Lucilia silvarum should be italicized

From Line 376 It seems that this information should go to the introduction, to introduce and inform readers about colonization of furred carcasses and to discuss the motivation and importance of the study. I believe the competition discussion instead, is quite useful as it contextualizes the importance of the findings of this manuscript

Line 431 Check grammar/spelling of this sentence

Author Response

Thank you to the reviewer for your thoughtful suggestions and feedback. We have made the changes requested and have listed the changes below:

Lines 40-42 – rephrased this sentence by breaking it down into multiple sentences.

Revised manuscript: Lines 40-43

Line 42 – explained the importance of development data.

Revised manuscript: Lines 43-49; Lines 49-56 also explain the importance of understanding the development of blow flies and their oviposition behavior.

History of L. silvarum:

  • Zumpt does not explain how Dunker identified silvarum in 1891.
  • Dodge could have used Hall 1948 key, but it does not explicitly say that is how Dodge ID’d the flies
  • Hanski does not state how the flies were ID’d
  • Haskell does not explicitly state how the flies were ID’d, but that they were ID’d and voucher specimens were sent to other calliphorid experts (Baumgartener, Hall).
  • Davies does not state how the flies were ID’d, just take they were ID’d as dried specimens and that females of caesar and L. illustris could not be separated due to the need of extruding the ovipositor as described by Spence 1954
  • Brothers also doesn’t explicitly say how the flies were ID’d, but does cite Hall 1948
  • Adair does not explain how the flies were identified, cites Hall 1948

The above mentions of possibly methods of identification were not added to the manuscript because they weren’t explicitly stated in source material.

  • Fremdt were identified the flies morphologically using Rognes 1991 and then molecularly by analyzing the mitochondrial cytochrome c oxidase I (COI)

Revised manuscript: Lines 231-233

Line 199 – Lucilia silvarum is italicized

Revised manuscript: Line 178

Figure 1 caption - Lucilia silvarum is italicized

Revised manuscript: Line 242

Line 294 - Lucilia silvarum is italicized

Revised manuscript: Line 330

Line 296 - Lucilia silvarum is italicized

Revised manuscript: Line 332

Methods

Line 312 – deleted sentence and explained materials/methods section in more detail and in a more cohesive way

Revised manuscript: Line 346-382

Line 323 - Included how many samplings occurs and on what experimental days

Revised manuscript: Line 369

Statistical Analysis

Included normality test

Revised manuscript: Line 385-386

Line 376 - Lucilia silvarum is italicized

Revised manuscript: Line 408-409

Information about colonization of furred carcasses (Lines 378-404) – Moved to introduction: Previous Studies Involving Cats

Revised manuscript: Lines 301-322

Line 431 – Reworded to fix the grammar of the sentence

Revised manuscript: Line 457-458

Thank you for your thoughtful consideration,

Kelly Bagsby

Reviewer 2 Report

Comments and Suggestions for Authors

The work should be revised to improve it in several ways. Some of them are formal aspects (but not less important), such as those concerning the References. The writing should be also revised to avoid unwanted repetitions or unnecesary statements. 

In the file you'll find some comments.

Author Response

Thank you to the reviewer for your thoughtful suggestions and feedback. We have made the changes requested and have listed the changes below:

Line 12: Changed “was known” to “has been referred”

Revised manuscript: Line 12

Line 16: Changed “was” to “appeared as”

Revised manuscript: Line 16

Line 45 – Reworded

Revised manuscript: Line 47

Line 51 – Added adults

Revised manuscript: Line 64

Line 52 – Changed wording to include higher and lower threshold temperatures affecting development

Revised manuscript: Line 65-67

Line 54 – Changed “and” to “that”

Revised manuscript: Line 68

Line 62 – Added more recent and relevant studies analyzing VOCs

Revised manuscript: Line 76

Lines 72-75 – Deleted these lines to reduce redundancies/reworded section

Revised manuscript: Line 49

Line 82 – Added maximum

Revised manuscript: Line 54

Line 89 – Added in North America

Revised manuscript: Line 91

Lines 106-107 – Cut out redundancies to make a more seamless sentence

Revised manuscript: Lines 110-112

Line 110 – The template is splitting up the word “myiasis”, with the revision it is fixed.

Revised manuscript: Line 113

Lines 111-113 – Reworded myiasis definition

Revised manuscript: Line 112-114

Lines 119-125 – Reworded and reorganized this section

Revised manuscript: Line 141-144; Line 146-147

Lines 130- 133 – Deleted results from current study from this section.

Lines 133 – 153 – Deleted redundant information and added % collected to the table so the readers can see the frequency of species collected at or near the research site.

Revised manuscript: Line 280-284; Table 3

Line 157 – Changed discovered to described

Revised manuscript: Line 156

Lines 166-186 – Moved the section of myiasis to the introduction after myiasis was introduced. Reorganized section as well to be more fluid.

Revised manuscript: Lines 115-152

Line 187 – Explaining the origins of blow flies parasitizing frogs is important regarding the history of Lucilia silvarum and makes most sense to be included in this section.

Revised manuscript: Lines 166-170

Line 193 – Added who and when; changed “thought” to “claimed”

Revised manuscript: Line 172

Line 199 – Italicized L. silvarum

Revised manuscript: Line 178-179

 Line 202 – Added citations for Jones et al, 2019 and Hall, 1948

Revised manuscript: Line 182 and 185

Lines 210-212 – Deleted parts and rephrased previous sentence to reduce redundancies

Revised manuscript: Lines 189-192

Line 217 – Deleted in Indiana

Revised manuscript: Line 197

Line 234 – Provided clarification to explain this sentence/rephrased sentence

Revised manuscript: Lines 215-220

Line 242 -243 – Deleted portion of sentence, added other species because only the larvae collected were L. silvarum, but the adults collected were Thanatophilus coloradensis and Calliphora coloradensis

Revised manuscript: Lines 226-228

Line 282 - 288 – Rephrased this section

Revised manuscript: Lines 226-271

Lines 285 – 288 – Deleted sentences

Lines 292 – 296 – Rephrased section to explain the purpose of this manuscript

Revised manuscript: Lines 323-333

Lines 304-306 – Added a sentence about how many light/dark cats there were

Revised manuscript: Line 342

Line 308-309 – Added environmental conditions of research site and ecoregion

Revised manuscript: Lines 350-352

Line 310-312 – Deleted sentence

Line 318 – Temperature data

Revised manuscript: Lines 361-362

Line 337 – Deleted citation for personal communication

Revised manuscript: Line 382

Line 361 – Added other areas that L. silvarum has rarely been collected from

Revised manuscript: Line 412-414

Line 364 – Lucilia coeruleiviridis did not survive so I can’t make any assumptions or conclusions about if it did survive.

Line 367 – See table 3 for data about L. silvarum collections

Line 378 – There are other papers, including table 1, that have more detail about other animals, but the focus of this study was on cats.

Lines 378 – 397 – moved to introduction: Previous Studies Involving Cats

Revised manuscript: Lines 301-322

Lines 415- 418 – This line is a statement introducing my ideas about L. silvarum preferring a lack of competition

Revised manuscript: Line 441-443

Lines 420-421 – Deleted redundancy

Revised manuscript: Lines 444-447

Line 425 – 430 – Explaining how and why L. silvarum may prefer carcasses with a lack of competition; explained in more detail

Revised manuscript: Lines 448-456

Line 431 – Added in Indiana

Revised manuscript: Line 457

Lines 436-438 – Summarizing this paper and the misidentifications of L. silvarum is an important part of this paper and its history

References

Updated all references

Thank you for your thoughtful consideration,

Kelly Bagsby

Reviewer 3 Report

Comments and Suggestions for Authors

This is a very interesting story on the impact identification errors can have on the understanding of a specific dipteran species. Furthermore, a small experiment illustrates the path to uncovering decades old misunderstandings.

The title suggests an original article about cats but large parts of the manuscript read like a review, as the text should do under its label. Please decide which path to take for this publication, as the current state is neither the one nor the other, and rewrite accordingly. If you consider a structure more pertinent to a review, it might be possible to add the experiment as a chapter. This route would also require a rewrite of the discussion as it does not match the rest of the manuscript in its current state.

Please remove Lucilia silvarum from keywords. Words from the title do not need to be included in keywords.

L. 126: Please revise this sentence.

The explanations of myiasis (from L. 166) should be taken out of the chapter “The History of Lucilia silvarum”. In exchange, the section first mentioning L. silvarum (from L. 118) should be included in the history chapter. Furthermore, the summary of Indiana studies might be better placed after the history chapter.

L. 366: Can you estimate the percentage of eggs collected which failed to develop in the laboratory?

Please check the manuscript for italic font in all instances of L. silvarum.

Author Response

Thank you to the reviewer for your thoughtful suggestions and feedback. We have made the changes requested and have listed the changes below:

Structure of manuscript – Changed article type to Technical Note

Revised manuscript: Line 1

Lucilia silvarum was removed from keywords

Revised manuscript: Line 37

Line 126 – Revised sentence

Revised manuscript: Line 276-277

Moved the explanation of myiasis (from Line 166) to the introduction when explaining which flies are involved with myiasis and what myiasis is.

Revised manuscript: Line 115

Moved first mention of L. silvarum (Line 118) to the history of L. silvarum

Revised manuscript: Line 163

Lines 126 – 155 – Moved the Indiana fly section after the history of Lucilia silvarum

Revised manuscript: Line 275-296

Line 366 – Added an estimate of the percentage of eggs that failed to develop

Revised manuscript: Line 419-420

Updated all instances of L. silvarum so they are italicized

Thank you for your thoughtful consideration,

Kelly Bagsby

Round 2

Reviewer 2 Report

Comments and Suggestions for Authors

The paper can be accepted in the present form

Author Response

Thank you to the reviewer for your thoughtful suggestions and feedback throughout this review process.

Kelly Bagsby

Reviewer 3 Report

Comments and Suggestions for Authors

The revised manuscript is more comprehensible than the first attempt. However, it is still too long. Many superfluous expressions should be eliminated.

A few examples, there are more which should all be addressed:

Lines 23-25: Three consecutive sentences start with Lucilia silvarum, consider combining them into one sentence.

Line 48:  “TOC estimates can be calculated using the development data from laboratory studies. Blow fly development data has been studied extensively in laboratory settings.”

Line 77: “As decomposition progresses, the remains are constantly changing and the tissues decompose.”

In Chapter “Previous Studies of Blow Flies in Indiana” (from Line 278) table 3 (should be table 2) could be omitted, as the detailed information can undoubtedly be found in the review by Hans et al. [98].

Line 341: “On the initial day of this study, the cat carcasses were photographed and initial documentation included the fur color and weight (kg). The cats had a mean weight of 5.3 +/- .11 kg. There were three light cats and six dark cats with a mean weight of 5.3 +/- .11 kg.”

Several repetitions in regards to content from the introduction can be found in the discussion section, please amend!

Further comments:

Table 1: Is there a reason for this table not to be in alphabetical order?

Line 158: remove “Table 2. cont.”

Line 234: please reorder - mention the cases first, identification methods after.

Line 383: is reference [117] correctly places here?

Line 411: check reference

Line 418: “Of the 38 samples collected in this study, 57% of the samples failed to develop.” Should be in the results section.

Author Response

Thank you to the reviewer for your thoughtful suggestions and feedback. We have made the changes requested and have listed the changes below:

Line 23-25 – Combined sentences to reduce redundancies.

Revised manuscript: Line 24-25

Line 48-50 – Combined/deleted sentences to reduce redundancies.

Revised manuscript: Line 48-49

Line 77-78 – Combined sentences to reduce redundancies.

Revised manuscript: Line 76-78

Line 89-90 – Deleted sentences to reduce redundancies.

Line 97-98 – Deleted part of this sentence.

Revised manuscript: Line 93-94

Table 1 – Alphabetized the table, bobcat is under cat and coyote is under dog because they are of the same genus (Felis and Canis, respectively)

Revised manuscript: Line 97-101

Line 110-116– Deleted and reworded sentences to reduce redundancies, deleted common names for blow flies that are not talked about in the following paragraphs, deleted sentences about myiasis by specific species because they do not add to the topic of this manuscript, deleted facultative myiasis sentences because they do not add to the topic of this manuscript.

Revised manuscript: Line 103-118

Line 158 – Deleted Table 2 cont.

Line 168 - Combined sentences to reduce redundancies.

Revised manuscript: Line 127-129

Line 170-171 – Reworded/deleted sentences to reduce redundancies.

Revised manuscript: Line 130

Lines 174-188 – Reorganized, combined, and deleted sentences to reduce redundancies.

Revised manuscript: Line 134-146

Line 197-199 – Deleted sentence to reduce redundancies.

Line 203-204 – Deleted sentence to reduce redundancies.

Line 207-211 – Combined and deleted sentences to reduce redundancies.

Revised manuscript: Line 163

Line 214-218 – Deleted and combined sentences to reduce redundancies.

Revised manuscript: Line 166-186

Lines 224-242 – Deleted sentences about cases involving humans because it is not related to animals with fur.

Figure 1 – Added more detailed descriptions and citations to caption since many sentences from the manuscript that originally cited these works were deleted

Revised manuscript: Line 175-193

Lines 257-259 – Combined sentence to reduce redundancies.

Revised manuscript: Line 200-202

Line 278-297 – Delete Previous Studies in Indiana and Table 3, added a sentence about Lucilia silvarum in Indiana, updated mentions of new species by including their full name and authority.

Revised manuscript: Line 221-222, line 226-235

Line 301-302 – Deleted sentence to reduce redundancies.

Line 335-383 – Reworded Materials and Methods section to reduce redundancies since we are citing the Bagsby et al (In Press) manuscript which explains these materials and methods in detail.

Revised manuscript: Line 256-291

Line 383 – Deleted personal communication citation.

Line 398-400 – Deleted Table 4, put relative abundances in results paragraph.

Revised manuscript: Line 302-303

Line 403-409 – Deleted and combined sentences to reduce redundancies.

Revised manuscript: Line 309-311

Line 411 – There was a missing comma, it is added now.

Revised manuscript: Line 312

Line 414– Combined sentences with line 422-423 to reduce redundancies.

Revised manuscript: Line 315-316

Line 418-419 – Moved sentence about samples to results.

Revised manuscript: Line 300-301

Lines 424-425 – Combined sentences to reduce redundancies.

Revised manuscript: Line 324-326

Line 449 – Deleted portion of sentence to reduce redundancies.

Revised manuscript: Line 347-349

Thank you for your thoughtful consideration,

Kelly Bagsby
